# A study to evaluate WASH interventions and risk factors of diarrhoea among children under five years, Anloga district, Ghana: A research protocol

**Delia Akosua Bandoh** *, **Ernest Kenu, Duah Dwomoh, Edwin Andrew Afari, Mawuli Dzodzomenyo**

University of Ghana School of Public Health, Accra, Ghana

* deliabandoh@gmail.com

## Abstract

### Introduction

Good Water, Sanitation and Hygiene (WASH) practices, introduction of Rotavirus vaccination, zinc supplementation and improved nutrition have contributed significantly to the reduction of diarrhoea morbidity and mortality globally by 50%. In spite of these gains, diarrhoea still remains a leading cause of morbidity and mortality in children under-five. Causes of diarrhoea are multifaceted with many factors such as seasonality, behaviour, pathogenicity, epidemiology, etc. However, assessments on the causes of diarrhoea have generally been tackled in silos over the years focusing only on particular causes. In this study, we describe an integrated approach (evaluating WASH interventions implantation processes, assessing epidemiolocal risk factors, and identifying pathogens causing diarrhoea) for assessing determinants of diarrhoea.

### Methods

The study has ethical approval from the Ghana Health Service Ethical Review Committee (GHSERC:020/07/22). It will employ three approaches; a process evaluation and a case-control study and laboratory analysis of diarrhoea samples. The process evaluation will assess the detailed procedures taken by the Anloga district to implement WASH interventions. A desk review and qualitative interviews with WASH stakeholders purposively sampled will be done. The evaluation will provide insight into bottlenecks in the implementation processes. Transcribed interviews will be analysed thematically and data triangulated with reviews. A 1:1 unmatched case-control study with 206 cases and 206 controls to determine risk factors associated with diarrhoea in children under-five will also be done. Odds ratios at 5.0% significance level would be calculated. Stool samples of cases will be taken and tested for diarrhoea pathogens using Standard ELISA and TAQMAN Array Card laboratory procedures.

**Data Availability Statement:** All relevant data from this study will be made available upon study completion.

**Funding:** The authors received no specific funding for this work.

**Competing interests:** The authors have declared that no competing interests exist.

## Expected outcome

It is expected that this framework proposed would become one of the robust approaches for assessing public health community interventions for diseases. Through the process evaluation, epidemiological case-control study and pathogen identification, we would be able to identify the gaps in the current diarrhoea assessments, come up with tailored recommendations considering the existing risk and assumptions and involve the relevant stakeholders in reducing the diarrhoea burden in a coastal setting in Ghana.

## Introduction

Water, Sanitation and Hygiene (WASH) interventions are a well-established interventions for reducing diarrhoea incidence. Over the years, WASH in addition to interventions such as introduction of rotavirus vaccination for infants, zinc supplementation and improved nutrition have contributed significantly to the reduction of diarrhoea morbidity and mortality globally [1–8]. From 1990 to 2015, diarrhoea morbidities have reduced by about 50% (from 1.8 million to 842,000) [9, 10]. In Ghana, diarrhoea deaths have reduced from over 26,000 in 2020 to about 6,000 in 2019 [11]. In spite of these gains made globally, diarrhoea still remains one of the leading causes of morbidity and mortality in children under five years [12, 13].

Generally, gains made in the decline of diarrhoea have been marginal globally showing geographical disparities [14]. Developing countries still bear the greatest burden of diarrhoea with lowest rates of decline recorded in Africa [14]. In 2019, Africa recorded 50% (370,000) of all diarrhoeal related diseases death among children under five years [15]. In Ghana, diarrhoea still remains part of the top ten causes of morbidity and mortality among children under five years [16]. In 2019, over 2,600 children under five died from diarrhoeal related causes in Ghana [11]. Yet diarrhoea is preventable.

Though the causes of diarrhoea are multifaceted and influenced by various factors such as seasonality, behaviour, pathogenicity, epidemiology, etc [17–19], interventions to reduce Ghana's diarrhoea burden have mainly focused on improving WASH practices. Provision of portable water and improved toilet facilities, handwashing stations and proper waste management have been the strategy for some decades [20]. Most of these interventions have been generalised across geographic areas with little consideration for the geographic differences that exist.

Similarly, assessments of diarrhoea have generally been done with the same approach. Researches on either observational studies on risk factor assessments, assessments on presence or absence of WASH intervention have mainly been the method for assessing diarrhoea causes [23–25]. A few have added laboratory investigations to determine the pathogens causing diarrhoea [21–25]. Since most of these assessments have not comprehensively covered all the causes of diarrhoea, conclusions from them drawn come with quite an amount of uncertainty.

Therefore, there remains a paucity of information on multifaceted assessments of diarrhoea which cover the details of WASH interventions, epidemiological risk factors and laboratory analysis. These kinds of assessments could provide a comprehensive picture on the problems and solutions to the diarrhoea menace.

For a low-middle-income settings like Ghana, where WASH implementation is one of the main interventions for diarrhoea prevention [1, 26], such a holistic assessment of the different drivers of the problem would help in identifying the gaps in the existing system and relevant stakeholders to help address them.

In this study, we described the need for an integrated approach (a process evaluation for implementing WASH interventions; an epidemiolocal case-control study, and an assessment of diarrhoea pathogens) for assessing determinants of diarrhoea among children under five years in a coastal community in Ghana using an adapted theory of change framework.

This new approach would help provide information on the gaps in the implementation of the existing WASH interventions, epidemiological risk factors, and the information on the type current type of diarrhoea causing pathogens (Fig 1).

## Materials and methods

### Aim and study design

In the current study, determinants of diarrhoea among children under five years, Anloga district, Ghana will be assessed using a mixed design approach and data collected by qualitative and quantitative methods. This approach will help in capturing all relevant information on the various factors leading to diarrhoea. The assessment will be done using three main approaches: a process evaluation of WASH interventions implemented in Anloga district, an epidemiological case-control study and a laboratory analysis of pathogens causing diarrhoea. The entire assessment is expected to be carried out over a 15-month period.

### Study setting

The study site will be Anloga district, a coastal district in the Volta region of Ghana. The district has a projected population of about 112,662. About 40% of the district land area is covered

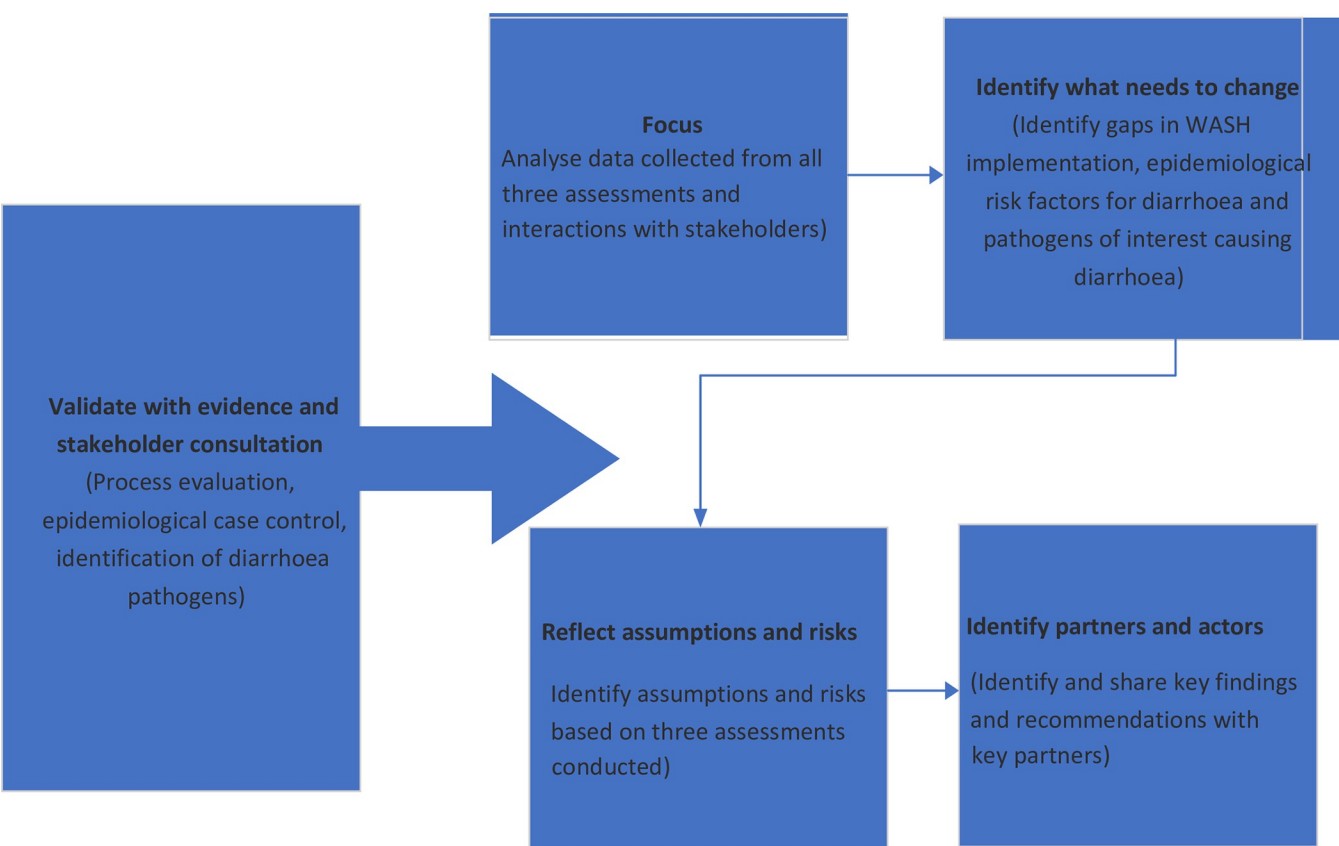

**Fig 1. Modified theory of change framework for the comprehensive assessment of diarrhoea cause.**

with water with 20% being wetlands [27]. The district is prone to flooding and experienced its most recent tidal flooding in March 2021. Where water households was destroyed and water sources were submerged in water [28].

The district is governed by a district chief executive in a district assembly which is the local government authority in charge of overall development of the district [29]. The assembly is made up of several units which manage and implement various health, environmental and educational interventions. Two communities in the district will be randomly selected models for assessing the processes taken to implement WASH in the district.

## Processes, interventions, and comparisons

Details on the three proposed approaches that will be used for the assessment are described below:

### Approach 1: Process evaluation of WASH intervention implementation

**Study population.** The population for the process evaluation would be stakeholders of WASH in the district. Namely; district assembly staff who play key roles in implementing and managing WASH activities, community opinion leaders and resident community members who have benefited from various WASH interventions in place.

To obtain the list of the district's main stakeholder groups or organisations involved in WASH intervention, an initial engagement has been held with the district assembly. Based on the interactions, the key groups with can be interviewed will be departments in the assembly directly involved in planning and implementing WASH activities in the district assembly, and communities in the district who have benefited from various WASH interventions.

### Sample size calculation

No sample size was calculated for the survey. All stakeholders playing a key role in the implementation process at the district and community level will be purposively identified and interviewed. However, based on initial enquiries with the district, the individuals in the table below will be engaged (Table 1):

In total, it is estimated that at least 34 people would be engaged. For the key informant interview, all the five heads of department in the district assembly involved in WASH implementation will be interviewed. In each community, a total of 5 leaders would be engaged in in-depth interviews. Two focus groups made up of at least 8 people will also be conducted in the same community. Finally, a checklist will be administered to fifteen individuals living close to WASH structures (Table 1).

### Sampling

**Sampling at the district level.** The individuals that will be engaged for the interviews will be selected based to their position and previous experiences with implementing WASH

**Table 1. Population to be sampled for qualitative interviews.**

| Type of interview | Estimated number of people to be interviewed | Level of interview |
| --- | --- | --- |
| Key informant interview (KII) | 5 people | District level |
| In-depth interview (IDI) | 5 people per community | Community |
| Focus group discussion (FGD) | 8–10 per discussion with 2 discussions per community | Community |
| Checklist | 5 people per WASH structure type (15 per community) | Community |
| **Total** | **Minimum of 40 people** | |

interventions. This would be heads of departments constituting the WASH team (the team in charge of planning and implementation of various WASH interventions). The departments will be Environmental Health, Planning, Works District Health Directorate and Social Welfare.

**Sampling at the community level.**   In each community, the community leader and the opinion leaders in charge of the specific types of WASH facilities for in-depth interviews will be identified. Namely the individual in charge of all water sources, individual in charge of all toilet facilities, individual in charge of waste dump sites in each of the two communities and the head of the health facility in the community.

To select participants for the focus group discussion, the WASH structures in each community will be identified and obtained information on when they were built. Of the WASH structures, the two most recent ones in the community as the start of the study will be selected. This will help in identifying the most recent structures which would be the reference point for assessing WASH implementation in the community. This reference point will be used enable community members remember with easy their experiences during the structure's development. In the next step, we will map out all households 200 meters around the WASH structure and randomly select eight to ten household heads in households from the location to take part in the focus group discussion. In total four focus group discussions will be held, two in each of the communities. A participant of the FGD would be a community member above 18 years, who is a household head and lives within 200meters of the selected WASH structures.

To further assess community members experiences with WASH intervention implementation, a checklist will be administered to all houses 100meters around the most recent WASH structure.

## Data collections

Qualitative methods will used for the WASH implementation process evaluation because it provides the opportunity to explore and understand the experiences of each stakeholder during the implementation.

## Desk review

We will scope for all documents on water and sanitation at the national and district levels. Two data extraction sheets will be used for the evaluation. The first sheet will collect information on the WASH interventions the districts implemented, targets for the district, interventions the district was able to implement. The second extraction form would collect data on national level policies, guidelines and strategies for implementing WASH interventions at different administrative levels in the country. There will be engagements with the district assembly to find out the documents they use in their activities and obtained copies of them. These will include the mid-term development plan of the district, annual plans, Community Water and Sanitation Agency Sector Guidelines–General (Rural Communities & Small Towns) and any other relevant documents available. From these documents, extract information on steps and processed for developing various kinds of water and sanitation interventions, stakeholders to be involved, targets set to develop WASH structures, targets achievement will be extracted. From the review, we will develop a framework that would be used in comparing the steps currently being used by the district.

**Key informant interviews (KII).**   The selected heads of department at the district level will be engaged individually in a KII to document their experiences and roles played in WASH interventions that have been implemented in the district. The interviews will be held at the district assembly. A trained research assistant will administer the interviews using a key

informant interview guide. For each interview, there will be a note taker, to record the interview and also take notes during the interview.

In-depth interviews

In the selected communities, the identified opinion leader, community member in charge of water structures, community member in charge of all sanitation structures and waste dump sites will be interviewed individually. Questions on their experiences with implementing WASH interventions in their community namely, the role they played, the steps they observed during the process, how they were engaged, management and sustainability of the intervention will be asked.

In-depth interviews (IDI) will be done in the community. A quiet spot in the community with no distractions will be identified with the help of the community leaders. Interviewers will be done by two trained research assistants. One will conduct the interviews using an IDI guide and the other take notes in addition to recording the interview.

**Focus group discussions.** A FGD with selected household head will be conducted in the community. Questions will cover how they were engaged, the roles they played, how the implementation was managed and the process of handing the structure over to the community.

Focus group discussions will be held close to the selected WASH structure reference point. The interview will be conducted by three research assistants, one moderator asking the questions, a note taker and one person supporting with organization and coordination of the discussion. Interviews will be conducted in the local dialect. Each discussion will last for a minimum of 30 minutes.

**Checklist.** A checklist on community awareness of the implementation will be administered by interviewing households living within the immediate surroundings of the identified WASH structure. The checklist will cover the status of the structure, information on how it came about, community's role in the process, and any known contributors to the implementation.

The checklist will be administered to five houses located 100meters around the WASH structure. One adult, preferably a household head would answer the checklist questions. In total, fifteen interviews will be held per community, five around the most recent water source, five around the most recent toilet structure and five around the most recent waste dump site. If a community does not have all three, the checklist would be administered around only the structures available. The variables included in the checklist will be; type of WASH structure, individual or organization who led development of the intervention, origin of WASH structure, donors of structure, information given on use of WASH structure.

## Data management and analysis

The data generated will be triangulated to produce a concise picture of the processes undertaken in WASH implementation and the likely bottlenecks. Information from desk review was extracted into a Microsoft Excel document and used in development of a WASH framework based on the national and desk documents. If a national framework already exists, it would be adapted and used as a guide to assess how implementation was done in the district. The framework will outline the required steps supposed to be taken in implementing a WASH intervention at the district level.

All interviews will be transcribed verbatim and analysed using model for inductive thematic analyses by Braun and Clarke [30] (This involved: (1) familiarisation with the data; (2) systematic data coding; (3) generating initial themes; (4) developing and reviewing themes; (5) refining, defining, and naming themes; and (6) writing the report.

Findings from the checklist will be summarised based on the variables collected to provide more detail on the views from the focus group discussions. Analysis for the qualitative interviews will be done using NVIVO Version 14.

Information from the qualitative report in addition to the checklist will be used to identify the actual steps in the WASH framework the district followed during their implementation. Reasons for non-compliance to any of the steps will also be obtained from the report. Through the comparison, researchers will identify the gaps, strengths and bottlenecks in the processes the district is using. Additionally, researchers will calculate the percentage achievement for WASH related interventions from the documents with their set targets and activities.

## Assessment 2: Epidemiological case-control study

**Study design.**   After the process evaluation, a case-control study will be conducted with children under five years to determine the epidemiological risk factors of diarrhoea in the district.

## Study site selection

All health facilities in sub-districts along the water bodies will be selected, these are Anloga, Anyanui and Tegbui. Of these three sub-districts, facilities which consistently recorded diarrhoea cases monthly in 2019 [31] will be selected as sites for data collection. The reference year of 2019 was chosen because it reflected the state of the facility before the COVID-19 pandemic occurred.

Using this criterion, the following facilities will serve as data collection sites: Anloga health center, Anyanui health center, Atokor health center, Kodzi health center, and Tegbi health center will be selected as sites for recruitment of cases.

## Study population

Children under five years in the Anloga district and their caregivers will be the target population for the study

## Sample size calculation

The sample size for the case-control study was calculated using the following assumptions abd parameters and an exposure of use of soap for hand washing at critical times among controls in a similar study done in Ethiopia was used for calculation of the sample size. Assumptions taken into consideration were Detectable odds ratio– 2; Exposure in control (use of soap for hand washing at critical times—74.1%; significance level—5%; Power (probability of detecting a real effect) - 80%; Control/case ratio– 1 [19, 32]. A minimum sample size of 206 cases and 206 controls will be recruited for the study. To document seasonality, recruitment would continue for a minimum of 12 months even if after 206 cases and controls have been recruitment.

## Sampling

Cases (children under-five years with their caregivers) will be recruited from five health facilities consecutively as the child is confirmed by the healthcare provider to have diarrhoea. This approach will be used to ensure that all children with diarrhoea reporting to the health facility are not missed. The caregivers will be referred to the research assistants by the health worker who attended to them. Selection of cases was done at the health facility to avoid case misclassification.

Recruitment of controls would be done in the community. For each case recruited, a corresponding control will be identified in the community the case lives. Using the residential information provided the case, a trained health worker from the facility would trace the home of the case. With that home as a landmark, a control would be selected from the next house. A control would be a child less than five years who has not been diagnosed with diarrhoea over the past seven days. If there is no child under five years in the next house, interviewer will move to the next house. This procedure will be repeated till they enter a house with a child under five years. The same selection criteria will be used across the five study sites.

## Case definition

For the case control study, the following case definitions based on the WHO standard definition for Diarrhoea [33] will be used.

A case in this study will be defined as a child under five years in Anloga district who is confirmed to have diarrhoea by the Physician Assistant or Nurse in charge at the health facility during the study period.

A control will be defined as a child under five years living in the same community as the case who has not reported to any health facility with diarrhoea in the past seven days. The seven day period will be used to reduce recall bias [34].

Inclusion criteria for the study will be any child under five years with the caregiver available in the study community would be included in the study. Any child how has not been in the study community for more than seven days prior to the interview will be excluded.

## Data collection

For both cases and controls, a trained health work will administer a quantitative interview gathering information on their demographics, health seeking behaviour, dietary pattern, household WASH practices. Under demographics, date of birth of child, place of residence, caregiver age, educational level of household head, level of education of care giver, wealth index will be collected. For health seeking practices; date of onset of diarrhoea, treatment given prior to reporting at health facility, rotavirus vaccination status of child, enrollment on a health insurance scheme will be collected. For the WASH practices data on water source, toilet facility usage, water storage practices, sharing of toilet facilities, frequency of hand washing with soap and water and disposal of child's stools. For dietary practices, caregivers will be asked to recall the foods the child had eaten over the past 24 hours and food groups eaten over the past 7 days.

Also, the weight and height of each child will be taken using standard anthropometric methods for children under five years [35–37]. Anthropometric measures (weight and height) will be taken twice by two trained health professionals and recorded on the questionnaires. It will be ensured that all children will be cladded in only underwear or light clothing during measurements.

Weight measurement: For children below 24 months, mothers will be made to stand on the scale bare footed after all heavy objects she was holding or adorned with had been collected. The scale will then be tarred and the child handed to the mother on the scale and the weight of the child taken. Children above 24 months will by themselves be made to stand on the scale and their feet positioned slightly apart. They will be asked to stand still and the reading was taken and recorded on the questionnaire.

Length: Recumbent length will be taken for children below 24 months and measured with the infantometer. The child will be gently placed on the infantometer with his/her head against the head board. The child's head will be held in place by cupping the ears. It will be ensured

that the vertical line formed from the ear canal to the lower border of the eye socket the child will be perpendicular to the horizontal board, forming the Frankfort vertical plane. The other health professional ensured that the child's trunk will be straight and flat on the board. The foot board will be gradually pushed to the feet of the child with the left hand whiles the right will be used to hold the legs together in place. The length will be recorded.

Height will be taken for the rest of the children above 24 months with a stadiometer. The child will be asked to stand on the footboard with their back against the back board. We will ensure that the back of their head, shoulder blade, back, buttocks calf and their heel touched the back board of the stadiometer. The head will be positioned so that the horizontal line connecting the upper ear opening and lower edge socket of the eye ball run parallel to the base board. This will form the Frankfort horizontal plane. The tummy will be pushed in gently to help the child to stand straight and the head board pressed firmly on the top of the head. The reading will then be taken and recorded.

To reduce recall bias during the case control study and to confirm information the caregiver provides on the child's health, researchers will cross check all details given from the child welfare record book. Additional for cases, information on the date of onset of diarrhoea, symptoms they were experiencing and any actions taken will be collected.

## Data management and analysis

Quantitative data will be collected with KoboCollect software using electronic tablets and uploaded daily, daily checks would be done on the data uploaded to ensure there are no missing gaps. The complete dataset will be downloaded in the Microsoft Excel CSV format and imported into STATA version 17 for cleaning and analysis at the end of the data collection.

Quantitative data will be tested for normality using the skewness and kurtosis, Shapiro-Wilk and Shapiro-Francia tests for normality. Frequencies and Relative Frequencies will be generated for categorical and means computed for continuous variables of both cases and controls. The relationship between diarrhoea morbidity and the exposure variables (health seeking behaviour, household WASH practices and dietary pattern and anthropometric measurements) will be assessed using binary logistic regression models with robust standard errors accounting for clustering at the community level. Magnitude of the relationship will be reported in terms of odds ratios with their corresponding 95% confidence interval. All statistical test of significance will be assessed at 5%.

## Approach 3: Identification of diarrhoea aetiology

**Study design.**   A cross-sectional survey will be conducted among the cases to determine the pathogens causing diarrhoea using the molecular methods (ELISA and Real Time PCR) in the laboratory.

## Study population

All cases whose caregivers will consent and provide stool samples for the survey would be eligible for the laboratory assessment.

## Sample size calculation

Using the prevalence of rotavirus in West Africa [38], and the number of cases to be recruited into the study, prevalence = 26.4% cases = 206, a minimum sample of 122 will be obtained for testing.

## Sampling

All 122 samples will be tested for rotavirus using the ELISA method. After this 50% of the samples (61) will be selected by randomly generated unique identification numbers for TAQMAN Array Card PCR tests the presence of other pathogens.

## Laboratory testing methods (pathogen identification)

All stool samples will be transported within 30 minutes to a central placed refrigerator and stored immediately at 2-4˚C. All samples will be transported to the Noguchi Memorial Institute for Medical Research (NMIMR) for analysis. In the laboratory, samples will be assessed for stool adequacy and integrity for the tests to be conducted. Stool samples would be tested for rotavirus antigen using a commercially available Enzyme-linked immunosorbent assay kit, ProSpecT® Rotavirus Microplate Assay which is based on detection of group specific antigen in group A rotaviruses. This assay kit will be chosen because it is a qualitative enzyme immunoassay for the detection of rotavirus in human faecal samples [39]. The manufacturer's procedure will be followed in preparation of the sample and conducting the tests [39].

For randomly selected samples, the TaqMan Gene Expression Array Cards and Plates for Real-Time PCR for the detection of multiple enteropathogens for a single specimen will be done to identify other pathogens causing diarrhoea. The TAQMAN Array Card procedure from Lappan and colleagues for conducting the laboratory tests and analysis will be adopted [40].

## Data management and analysis

Data generated from the samples was exported in Microsoft Excel 2016 and cleaned. Frequencies of pathogens would be generated and presented in tables.

**Measurement of outcomes for three approaches.** There will be three main outcomes for this study. For the process evaluation, steps taking to implement WASH interventions and bottlenecks will be documented.

The outcome for the case control study would be the occurrence of diarrhoea in a child under five years and risk factors associated with diarrhoea.

The outcome for the cross-sectional study will be pathogens causing diarrhoea in the children under five years.

**Quality assurance.** A three-day training will be organized for data collectors before all study data will be collected. Data collectors will be university graduates with at least one year experience in collecting data for public health surveys. All the questionnaires were pre-tested in a coastal community in the greater Accra region. The pre-test will be done in a day. For the KIIs, a district assembly official in the community will be identified and interviewed. For the IDIs, two community opinion leaders will be interviewed, for the FGD, the community representative will be engaged to mobilize household heads for the discussion to be done. About 5 households were also interviewed with the checklist. Additionally, five caregivers will be interviewed to pre-test the case control study.

For the measurement tools, the weighing scales will be sent to the Ghana Standards Authority for calibration before field work begins. A professional nutritionist with experience in taking anthropometric measures during nutritional surveys will train the health professionals in each facility on how to take weight and height measurements.

In the laboratory, all laboratory standard operating procedures will be followed and personnel conducting the tests taken through a quality assurance training before the test are done.

**Ethical considerations and declarations.** The proposal for this study has be submitted to the Ghana Health Service Ethics Review Committee and received approval (GHSERC:020/07/

22). The district health directorate and the District Assembly have granted approval/permission for the study to be conducted in the district. Permission has been obtained from the community leaders and health facility head. The purpose of the study will be explained to all participants in detail and their questions answered before enrolment. Participants will be assured of confidentiality. Informed consent forms will be administered to all study participants prior to participation. Participation in the study will be voluntary and subjects will be informed that they can withdraw at any time they wish even after consent has been given and during participating in the study. Data identifying participants such as place of residence will be collected only during the consenting process and will not be linked at any point to the interviews conducted. Participants will be assigned codes after consenting and codes will used throughout the study process. Filled consent forms will not be accessible to authors. All information obtained from the study will be kept confidential on password protected computers. The study population is made up of children under five but those to be interviewed will be mothers/caretakers of the children. Given that children under five are below the age of assenting, mothers/ caregivers will give consent for the child to be part of the study. Data generated from the study would be shared without identifiers to relevant stakeholders for necessary action to be taken in reducing diarrhoea burden.

## Expected study outcome

The outcome of the study would be a framework with a comprehensive approach for assessing diarrhoea causes in various settings. Mainly by:

1. Identify the strengths and bottle necks in the WASH implementation processes carried out in the Anloga district

2. Provide evidence of the epidemiological risk factors related to diarrhoea in children under five in the study district

3. Document the types of pathogens causing diarrhoea in the study district.

With this framework, a broader picture on the interlinkages between the causes of diarrhoea can be identified and specific solutions to tackle each area designed for different settings. Additionally, gaps in existing diarrhoea assessments can be identified, and tailored recommendations considering the existing risk and assumptions developed. All this would be with the involvement of relevant stakeholders in reducing the diarrhoea burden in a coastal setting in Ghana and other places in the country with similar settings.

**Status and timeline of the study.** The study will run from October 2022 till October 2023. The process evaluation was conducted in October 2022. The case-control study began in November 2022. Participants are being recruited as and when diarrhoea cases report to the two health facilities. Recruitment would continue till October 2023.

## Discussion

## Other aspects not covered by proposal

Over the years, researchers have tackled various health problems in parts, focusing on a narrow perspective. Recent years have shown the need for multisectoral collaboration to provide a more holistic picture of the issue at hand. Responding to the COVID-19 pandemic is a clear example of how solving a problem with a multisectoral approach yields results. As the world gradually moves from solving problems in silos to using multisectoral approaches, considering an assessment which encompasses the various interventions of diarrhoea embraces new norm.

In this current method, the assessment covers both the implementers and the end users of the WASH interventions, epidemiological factors within the target population which could have been missed and provided information on the pathogens of interest.

Though one might argue that the epidemiological factors and pathogens of interest have been widely research, the emergence of climatic change and its consequences have been found to influence a number of outcomes including diarrhoea which is regarded as a climate sensitive disease. Thus, period studies on these areas in these times are helpful for understanding the changing patterns and the areas of interventions that need to be modified according as we build climate resilience.

The study seeks to describe a comprehensive framework for assessment of diarrhoea. This study would be carried out over a 15-month period to be able to capture any season changes in diarrhoea that occur throughout the year. To ensure that that is possible all diarrhoea cases reported in the health facilities from November 2022 to December 2023 (15 months) would be eligible to be part of the study.

This study is expected to identify bottlenecks in the processes used to implement WASH interventions implemented in the district through the process evaluation conducted. (remember accessibility and use) Also, the factors associated with diarrhoea in the coastal communities and the etiology of diarrhoea throughout the year will be identified. This information can help in identifying steps and processes to be taken to reduce the diarrhoea burden in different settings using existing approaches and help them to build resilience in the face of the detrimental effects for climate change.

## Limitations of the study design

A limitation to this study will be recall bias. Where case care givers are able to clearly remember events such as dietary and WASH practice and other events preceding the diarrhoea incident whiles controls may not able to do so entirely. To reduce this bias, recall for dietary pattern would start from the past 24 hours and for WASH practices, interviews would be done in the household with references made to items around them.

## Dissemination plans

The study findings will be submitted to the University of Ghana School of Public Health as a thesis report that would be available to the entire university community. Dissemination meetings will be held with the district, communities and health facilities. Findings will also be disseminated through publications and presentations at international conferences.

## Amendment and terminations

Given the effect of covid-19 pandemic on the healthcare system such as increase in hand washing practices, reduction in hospital attendance reported in various settings [41, 42], we would review the number of cases by the third month and compare with the number of cases recorded from previous years. if it is less, other facilities which record diarrhoea cases based on the district health records would be added to the study sites. All amendments to the protocol would be submitted to the Ghana Health Service ethics Committee for approval before implementation.

## Acknowledgments

We would like to acknowledge the Management and of the C2R-CD Project, for supporting the data collection process. We would also like to acknowledge the District health assembly

staff, health facility heads and staff, research assistants and project supervisor for their various contributions towards the ongoing project.

## Author Contributions

**Conceptualization:** Delia Akosua Bandoh, Ernest Kenu, Duah Dwomoh, Edwin Andrew Afari, Mawuli Dzodzomenyo.

**Data curation:** Delia Akosua Bandoh, Ernest Kenu.

**Formal analysis:** Delia Akosua Bandoh, Ernest Kenu, Duah Dwomoh.

**Funding acquisition:** Ernest Kenu.

**Investigation:** Delia Akosua Bandoh, Ernest Kenu, Duah Dwomoh.

**Methodology:** Delia Akosua Bandoh, Ernest Kenu, Duah Dwomoh, Edwin Andrew Afari, Mawuli Dzodzomenyo.

**Project administration:** Delia Akosua Bandoh.

**Resources:** Delia Akosua Bandoh, Ernest Kenu.

**Software:** Delia Akosua Bandoh, Ernest Kenu, Duah Dwomoh.

**Supervision:** Delia Akosua Bandoh, Duah Dwomoh, Edwin Andrew Afari.

**Validation:** Delia Akosua Bandoh.

**Visualization:** Delia Akosua Bandoh.

**Writing – original draft:** Delia Akosua Bandoh, Ernest Kenu, Duah Dwomoh, Edwin Andrew Afari, Mawuli Dzodzomenyo.

**Writing – review & editing:** Delia Akosua Bandoh, Ernest Kenu, Duah Dwomoh, Edwin Andrew Afari, Mawuli Dzodzomenyo.

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
