## [Decision Letter · Decision Letter 0]

12 Sep 2023

PONE-D-23-11119A study to evaluate WASH interventions and risk factors of diarrhoea among children under five years, Anloga district, Ghana: a research protocolPLOS ONE

Dear Dr. Bandoh,

Thank you for submitting your manuscript to PLOS ONE. After careful consideration, we feel that it has merit but does not fully meet PLOS ONE’s publication criteria as it currently stands. Therefore, we invite you to submit a revised version of the manuscript that addresses the points raised during the review process.

We look forward to receiving your revised manuscript.

Kind regards,

Furqan Kabir

Academic Editor

PLOS ONE

2. Please amend the manuscript submission data (via Edit Submission) to include author Edwin Andrew Afari.

Additional Editor Comments:

Please review reviewer' comments and address.

Reviewers' comments:

Reviewer's Responses to Questions

**Comments to the Author**

1. Does the manuscript provide a valid rationale for the proposed study, with clearly identified and justified research questions?

Reviewer #1: No

2. Is the protocol technically sound and planned in a manner that will lead to a meaningful outcome and allow testing the stated hypotheses?

Reviewer #1: Partly

3. Is the methodology feasible and described in sufficient detail to allow the work to be replicable?

Reviewer #1: No

4. Have the authors described where all data underlying the findings will be made available when the study is complete?

Reviewer #1: No

5. Is the manuscript presented in an intelligible fashion and written in standard English?

Reviewer #1: No

6. Review Comments to the Author

You may also provide optional suggestions and comments to authors that they might find helpful in planning their study.

Reviewer #1: I would like to acknowledge authors for planning to conduct this study. I have the following concerns and comments.

Minor concerns.

1. There are number of writing problems in the paper, which are hard to understand, which require proofreading.

2. The whole paper needs to be re-written and well-structured to improve the flow of information for one section to the other.

Major concern

1. The reason for conducting this study is not well justified.

2. Expected outcomes are not clearly written.

3. Reference usage needs to be fixed; there are sentences require citation- for example "About half of the 370,000 children under five years who died from diarrhea related diseases in 2019 occurred in Africa."

4. The design and how the studies are linked are not clearly presented.

5. There should be some sort of theoretical framework for evaluation study and can stand alone.

6. Page 7 line "The aim of this review would be to identify the standard process for WASH intervention implementation in the district"- I would have thought the WASH intervention has its own standard procedure and guideline and not clear how and why this aim is important.

7. The method for qualitative study lucks minimum sample size to be included in the study?

8. Selecting cases from health facility and controls from community would introduce some sort of selection bias.

9. Sample selection method for the case seems non-probabilistic (convenient sampling) and for the controls is not totally clear.

10. There is mix-up of method presentation: the section "Process, intervention, and comparison" is misplaced, not clear, and unaimed.

11. Some measurements are not clearly stated- for example "The weight and height of the child will be taken using standard WHO methods. "- does not give clue what the WHO standard is?

12. What aspect of the paper will be benefited from pathogen identification information and how is its implication rated?

7. PLOS authors have the option to publish the peer review history of their article (what does this mean?). If published, this will include your full peer review and any attached files.

Reviewer #1: **Yes: **Abel Dadi

---

## [Author Response · Author response to Decision Letter 0]

16 Nov 2023

Reviewer's Responses to Questions

Dear Reviewer, 

Thank you very much for your comments such have helped authors improve about the development and documentation of our proposal. 

We have below point by point responses to comments raised and clarifications on some of the issues that were raised. 

Thank you very much 

Yours sincerely 

Delia Bandoh 

( Corresponding Author)

Review Comments to the Author

You may also provide optional suggestions and comments to authors that they might find helpful in planning their study.

Reviewer #1: I would like to acknowledge authors for planning to conduct this study. I have the following concerns and comments.

Minor concerns.

1. There are number of writing problems in the paper, which are hard to understand, which require proofreading.

Response: 

Thank you for your comments. The entire paper has been rewritten, reformatted and proofreading done. 

2. The whole paper needs to be re-written and well-structured to improve the flow of information for one section to the other.

Response: 

Thank you for your comments. The entire paper has been rewritten and reformatted for clarity. 

Major concern

1. The reason for conducting this study is not well justified.

Response: 

The justification has been rewritten in the introduction. Pages 4-5

2. Expected outcomes are not clearly written.

Response: 

Expected outcome has been outlines on page 21. 

3. Reference usage needs to be fixed; there are sentences require citation- for example "About half of the 370,000 children under five years who died from diarrhea related diseases in 2019 occurred in Africa."

Response:

Referencing for the entire manuscript has been redone

4. The design and how the studies are linked are not clearly presented.

Response: 

The design for each of the three approaches has been presented separated to improve clarity. Pages 7 - 20

5. There should be some sort of theoretical framework for evaluation study and can stand alone.

Response: 

The framework of the evaluation study was adapted from the UNDP theory of change framework (https://unsdg.un.org/sites/default/files/UNDG-UNDAF-Companion-Pieces-7-Theory-of-Change.pdf), 

6. Page 7 line "The aim of this review would be to identify the standard process for WASH intervention implementation in the district"- I would have thought the WASH intervention has its own standard procedure and guideline and not clear how and why this aim is important.

Response: 

WASH intervention implementation varies based on the existing programme in place in the site. Over the years, the procedure used has gone through context specific modifications and adaptations. Thus, to carry out an evaluation, there is the need to know the exact process each setting has adapted.

7. The method for qualitative study lucks minimum sample size to be included in the study?

Response: 

Thank you for your comment. However, we beg to differ. The qualitative part of the study covered all the major stakeholders. They would be purposively selected based on their roles in WASH interventions in the setting. Technically, all relevant people needed are being interviewed. Thus, we do not agree with this comment.

8. Selecting cases from health facility and controls from community would introduce some sort of selection bias.

Response:

Thank you for your comment, however, we share a different opinion on this. Case and controls selected from the same locality would rather to prevent bias selection bias since cases and controls are being drawn from the same population. By selecting cases and controls from the same locality in a case-control study, we would be able to reduce biased estimates of the association between the exposures and diarrhoea. 

9. Sample selection method for the case seems non-probabilistic (convenient sampling) and for the controls is not totally clear.

Response: 

Thank you for your comment, selection is not convenient sampling but based on approaches used by another study (Breurec S, Vanel N, Bata P, Chartier L, Farra A, Favennec L, et al. (2016) Etiology and Epidemiology of Diarrhea in Hospitalized Children from Low Income Country: A Matched Case-Control Study in Central African Republic. PLoS Negl Trop Dis 10(1): e0004283. doi:10.1371/journal. pntd.0004283).

Selection of cases was consecutive, every single child diagnosed of diarrhoea in the study sites will be eligible to be part of the study till the sample size is obtained. 

10. There is mix-up of method presentation: the section "Process, intervention, and comparison" is misplaced, not clear, and unaimed.

Response: 

The methods section has been rewritten by approaches now for clarity. Page 7 - 20 

11. Some measurements are not clearly stated- for example "The weight and height of the child will be taken using standard WHO methods. "- does not give clue what the WHO standard is?

Response: 

The detailed procedure for anthropometric measurement has been added to the methods section. Pages 15 -16

12. What aspect of the paper will be benefited from pathogen identification information and how is its implication rated?

Response:

Information on the pathogens would complement the case control study. This is because the case-control study would identify any risk factors including rotavirus vaccination status, and WASH which can all be influenced by different types of pathogens.

---

## [Decision Letter · Decision Letter 1]

31 Jan 2024

PONE-D-23-11119R1A study to evaluate WASH interventions and risk factors of diarrhoea among children under five years, Anloga district, Ghana: a research protocolPLOS ONE

Dear Dr. Bandoh,

Thank you for submitting your manuscript to PLOS ONE. After careful consideration, we feel that it has merit but does not fully meet PLOS ONE’s publication criteria as it currently stands. Therefore, we invite you to submit a revised version of the manuscript that addresses the points raised during the review process.

We look forward to receiving your revised manuscript.

Kind regards,

Khin Thet Wai, MBBS, MPH, MA

Academic Editor

PLOS ONE

Additional Editor Comments:

This is the manuscript that elucidates the critical role of WASH interventions to decrease under five diarrhoea.

Please revise in line with reviewers' comments to strengthen scientific integrity.

Reviewers' comments:

Reviewer's Responses to Questions

**Comments to the Author**

1. Does the manuscript provide a valid rationale for the proposed study, with clearly identified and justified research questions?

Reviewer #2: Partly

Reviewer #3: Yes

2. Is the protocol technically sound and planned in a manner that will lead to a meaningful outcome and allow testing the stated hypotheses?

Reviewer #2: Partly

Reviewer #3: Yes

3. Is the methodology feasible and described in sufficient detail to allow the work to be replicable?

Reviewer #2: Yes

Reviewer #3: No

4. Have the authors described where all data underlying the findings will be made available when the study is complete?

Reviewer #2: Yes

Reviewer #3: Yes

5. Is the manuscript presented in an intelligible fashion and written in standard English?

Reviewer #2: No

Reviewer #3: Yes

6. Review Comments to the Author

You may also provide optional suggestions and comments to authors that they might find helpful in planning their study.

Reviewer #2: I do acknowledge the authors for conducting this study and the followings are my comments.

1. “WASH” is the acronym of Water, Sanitation and Hygiene. Please rearrange it in abstract and introduction.

2. For process evaluation of WASH intervention and implementation, you need to mention how many stakeholders will be included in KII and how many participants do you plan to recruit for IDI in community. As it is the qualitative approach, you don’t need to calculate sample size but need to be declared estimated number.

3. Please mention the participant selection criteria for FGD since you only mentioned household heads near the most recently constructed WASH infrastructures. You plan to conduct 2 FGDs for each community, what will be characteristics of the participants to assign in each FGD?

4. Please describe the data collection method for KII, IDI, FGD in detail (who will be interviewer? where will be conducted? What tools are you going to used? Who will be facilitator and note taker etc.)

5. What variables are included in checklist of WASH structure status and how many participants are you going to ask?

6. Data management and analysis for approach (1) is still unclear. Need to describe how to compare the results in detail to identify gaps. (WASH framework, data from qualitative approach and data from checklist)

7. For case-control study, if there is no under 5 children in the next house of the case, how will you select the control? Please mention it as well.

8. Have you pretested the questionnaire for both qualitative and quantitative study? If so, how did you conduct those? (where and with whom)

9. Have you validated the questionnaire? If so, please mention in detail.

10. Need to mention who will be the interviewers for case-control study.

11. Need to mention how to calibrate the weighing machines. Who will train 2 healthcare professionals for taking height and weight measurements?

12. Please also mention how to control interobserver variations for measurements.

13. Please add intext citation and reference for sample size calculation of diarrhoea aetiology.

14. You’ve mentioned only 94 stool samples are going to proceed for lab analysis. It means that the rest of the collected samples will be discarded? If so, please reconsider that matter as the collection of stool sample is not easy and simple.

15. The outcome of case-control study isn’t the occurrence of diarrhoea. Please revise the measurement of outcomes. Need to mention the measurement for diarrhoea aetiology as well.

16. There are 2 study population for case-control study (children and mothers/caregivers), you should obtain both assent and consent form.

17. Need to revise the references inline with guidelines of PLOS ONE.

18. English writing still needs to be improved. Thorough proofreading is highly recommended.

Reviewer #3: Reviewer comments

Abstract

Overall: the Abstract is clear and ok, but some few details are required.

Introduction: You mentioned “decline” – can give us the percentage.

Methodology: Included the analysis that will be done in the evaluation (thematic analysis) and the sampling technique for the part of the research. Include that odds ratio will be obtained at 5% significant level.

Expected outcomes: Let us stop using the word “we”- refer yourself or the people as researchers.

Introduction

We need to see more stats in the 1st part of introduction. Let the stats follow from a global perspective to sub-Saharan region to Ghana with regards to childhood diarrhoeal illness.

Also, some stats on WASH intervention should be added in the introduction to give a clear picture why the formulated – connect it to Ghana.

Make your rational or problem statement or justification clear in the introduction because it seems that you are still narrating about diarrhoeal and WASH interventions of the country if there are close to achieving it…etc

Fix some of the grammars and add some references like in Line 10. Line 9-12 – needs to be restructured.

Methodology

Aim & Study design: Restructure the first line. Give the rational for using mixed design approach.

Study setting: Fix the first sentence and add the exact location of the study setting (East of Volta region) and add square kilometre of the area. Add a line of the main activities that are done in that region. They can act as the rational for selection of the region as the study site. Mention the communities and am sure they have names. Let the information follow.

Process evaluation: Why did you select qualitative methods (add a rational) and why purposively sampling? Are we interviewing only people that attended the meeting? Make it clear. Just state that …meeting was held on WASH interventions and potential stakeholders that participated will be purposively selected for an interview”.

Characteristics of population and sample selection: Can you justify each sampling method that was used.

Approach 1: Names some stake holders that we are likely to have for evaluation. I think most of information is confusing under some paragraphs. Try and put sub-titles and avoid adding information in wrong sub-titles.

Sampling size calculation: At least give us an estimation of how many participants you like to include by looking at literature.

Sampling: Be clear on the sampling technique and give its rational or since you are doing a lot of sampling methods you can state a multi-stage sampling technique and explain random sampling and purposively sampling. Follow of points is required.

Data collection: Desk reviews (state the data extraction tool will be used) for interviews and FGD (How long will they last, in translation of them??).

Data management & Analysis: State the package that you will be used for analysis for both qualitative and quantitative

7. PLOS authors have the option to publish the peer review history of their article (what does this mean?). If published, this will include your full peer review and any attached files.

Reviewer #2: No

Reviewer #3: No

---

## [Author Response · Author response to Decision Letter 1]

17 Mar 2024

Topic: A study to evaluate WASH interventions and risk factors of diarrhoea among children under five years, Anloga district, Ghana: a research protocol

Response to reviewers comments 

Dear Editor, 

Thank you for getting providing expert reviewers for our protocol. Their comments have helped in shaping our protocol. 

Kindly see below a point-by-point response to the comments they raised for your perusal. 

Thank you very much 

Yours sincerely 

Delia Bandoh 

(Corresponding author)

Reviewers' comments:

Reviewer's Responses to Questions

Comments to the Author

1. Does the manuscript provide a valid rationale for the proposed study, with clearly identified and justified research questions?

Reviewer #2: Partly

Reviewer #3: Yes

2. Is the protocol technically sound and planned in a manner that will lead to a meaningful outcome and allow testing the stated hypotheses?

Reviewer #2: Partly

Reviewer #3: Yes

3. Is the methodology feasible and described in sufficient detail to allow the work to be replicable?

Reviewer #2: Yes

Reviewer #3: No

4. Have the authors described where all data underlying the findings will be made available when the study is complete?

Reviewer #2: Yes

Reviewer #3: Yes

5. Is the manuscript presented in an intelligible fashion and written in standard English?

Reviewer #2: No

Reviewer #3: Yes

6. Review Comments to the Author

You may also provide optional suggestions and comments to authors that they might find helpful in planning their study.

Reviewer #2: I do acknowledge the authors for conducting this study and the followings are my comments.

1. “WASH” is the acronym of Water, Sanitation and Hygiene. Please rearrange it in abstract and introduction. 

Response 

Changed, thank you for pointing it out. (page 2 and page 4). 

2. For process evaluation of WASH intervention and implementation, you need to mention how many stakeholders will be included in KII and how many participants do you plan to recruit for IDI in community. As it is the qualitative approach, you don’t need to calculate sample size but need to be declared estimated number.

Responses

The estimated number has been added on page 9, methods section.

About 40 people would be involved in the interviews at both the district and community levels. A summary table has been added to estimate the number that would be involved in the qualitative interviews.

Type of interview Estimated number of people to be interviewed Level of interview 

Key informant interview 5 people District level

In-depth interview 5 per community (10 people) Community 

Focus group discussion 8 – 10 per discussion with 2 discussions per community (40 people) community

Checklist 5 people per WASH structure type (15 per community) Community

Total 40 people 

3. Please mention the participant selection criteria for FGD since you only mentioned household heads near the most recently constructed WASH infrastructures. You plan to conduct 2 FGDs for each community, what will be characteristics of the participants to assign in each FGD?

Responses 

The characteristics has been added. Page 11, methods section.

For the focus group discussion, a community member above 18 years, who is a household head and lives within 200meters of the selected WASH structures will be eligible participants.

4. Please describe the data collection method for KII, IDI, FGD in detail (who will be interviewer? where will be conducted? What tools are you going to used? Who will be facilitator and note taker etc.)

Response 

Thank you for your comment. The roles have been tabulated below for ease of presentation. 

Type of interview Interview location Venue Personnel involved Tools 

Key informant interview District Assembly meeting room / offices Research assistant conducting interview and note taker KII guide

In-depth Interview Community meeting places Research assistant conducting interview and note taker IDI guide

Focus Group Discussion Community meeting places Research assistant moderating discussion, note taker, and research assistant supporting moderator FGD guide

Checklist Household Research assistants Checklist questionnaire 

The description been further explained on pages 12 and 13, methods section.For KII, the interviews would be held at the district assembly. A trained research assistant will administer the interviews using a key informant interview guide. For each interview, there would be a note taker, to record the interview and also take notes during the interview. 

For the IDI, interviews would be done in the community. A quiet spot of location in the community with no distractions would be identified with the help of the community leaders. Interviewers will be done by two trained research assistants. One would conduct the interviews using an IDI guide and the other take notes in addition to recording the interview.

Focus group discussions will be held close to the selected WASH structure. The interview will be conducted by three research assistants, one will serve as the moderator asking the questions, the second, a note taker and the last person would handle the recording of the discussion.

5. What variables are included in checklist of WASH structure status and how many participants are you going to ask?

Response 

Thank you for your comment. Added on page 13, methods section 

The checklist will be administered to five houses 100meters located around the WASH structure. One adult, preferably a household head will answer the checklist questions. In total, fifteen interviews per community, five around the most recent water source, five around the most recent toilet structure and five around the most recent waste dump site. If a community does not have all three, the checklist would be admisitrered around only the structures available. The variables included in the checklist will be; Type of WASH structure, usage of WASH structure, origin of WASH structure, donors of structure, information given on WASH structure

6. Data management and analysis for approach (1) is still unclear. Need to describe how to compare the results in detail to identify gaps. (WASH framework, data from qualitative approach and data from checklist)

Response

The section has been rewritten for clarity, methods section, page 14

The data generated will be triangulated to produce a concise picture of the processes undertaken in WASH implementation and the likely bottlenecks. Information from desk review will be extracted into a Microsoft Excel document and used in development of a WASH framework based on the national and desk documents. If a national framework already exists, it would be adapted and used as a guide to assess how implementation was done in the district. The framework will outline the required steps supposed to be taken in implementing a WASH intervention at the district level. 

All interviews (KII, FGI and IDI) will be transcribed verbatim and analysed using model for inductive thematic analyses by Braun and Clarke [27] (This will involve: (1) familiarisation with the data; (2) systematic data coding; (3) generating initial themes; (4) developing and reviewing themes; (5) refining, defining, and naming themes; and (6) writing the report. 

Findings from the checklist will be summarised based on the variables collected to provide more detail on the views from the focus group discussions. Information from the qualitative report in addition to the checklist will be used to identify the actual steps in the WASH framework the district followed during their implementation. Reasons for non-compliance to any of the steps will also be obtained from the report. Through the comparison, researchers would to identify the gaps, strengths and bottlenecks in the processes the district is using. This would help in identify the gaps in implementation. Additionally, researchers would calculate the percentage achievement for WASH structures from the documents with their set targets and activities.

7. For case-control study, if there is no under 5 children in the next house of the case, how will you select the control? Please mention it as well.

Response 

Thank you for your comment, this has been updated in the methods. page 17

If there is no child under five years in the next house, interviewer will move to the next house. This procedure will be repeated till they enter a house with a child under five years

8. Have you pretested the questionnaire for both qualitative and quantitative study? If so, how did you conduct those? (where and with whom)

Response: 

Yes, all the questionnaires were pre-tested in a coastal community in the greater Accra region. the pre-test was done in a day. For the KIIs, a district assembly official in the community was identified and interviewed. For the IDIs, two community opinion leaders were interviewed, for the FGD, the community representative mobilized household heads for the discussion to be done. About 5 households were also interviewed with the checklist. Additionally, five caregivers will be interviewed to pre-test the case control study.

A section on quality assurance has been added to the manuscript, methods section, page 23. 

9. Have you validated the questionnaire? If so, please mention in detail.

Response: 

Yes, questionnaire was validated. Prior to the pre-test, the questionnaire was developed by the lead researcher. The tools for the process evaluation were taken through content validation by two research team members who are experts in Monitoring and Evaluation assessments. For the case control tools, an epidemiologist, a statistician and an expert in WASH on the research team did a content validation to ensure the questions captured the construct of the research being undertaken. A face validation of all the tools was done during the pre-test to find out the ease of understanding of the questionnaire.

10. Need to mention who will be the interviewers for case-control study.

Response: 

Trained health workers were the interviewers for the case control study. This has been added in the methods, page 17. 

11. Need to mention how to calibrate the weighing machines. Who will train 2 healthcare professionals for taking height and weight measurements?

Response: 

Information has been added to the methods too. Page 23.

The weighing scales will be sent to the Ghana Standards Authority for calibration before field work begins. A professional nutritionist with experience in taking anthropometric measures during nutritional surveys will train the health professionals in each facility on how to take weight and height measurements. 

12. Please also mention how to control interobserver variations for measurements.

Response: 

To reduce interobserver variations for measurements, the following steps will be taken: Experienced data collectors will be trained by a professional nutritionist to follow the standard steps for taking each measurement. Additionally, data collectors will be given the opportunity to practice taking each measurement a number of times whiles being observed and their measurements recorded each tile till the values obtained are consistent. Finally, Standard Operating Procedure (SOP) documents will be given to data collectors to guide their activities and periodic monitoring activities involved monitoring the measurement procedures will also be done. 

13. Please add intext citation and reference for sample size calculation of diarrhoea aetiology.

Response: 

This has been added. Thank you, Page 22, sample size calculation for approach 3

Using the prevalence of rotavirus in Africa post rotavirus vaccination era [35], and the number of cases to be recruited into the study, Prevalence = 26.4% cases = 171, a minimum sample of 109 will be obtained .

14. You’ve mentioned only 94 stool samples are going to proceed for lab analysis. It means that the rest of the collected samples will be discarded? If so, please reconsider that matter as the collection of stool sample is not easy and simple.

Response: 

Thank you very much for your suggestion, all samples will be tested for rotavirus using ELISA technology. This would be the minimum test per sample. Some would then be randomly sampled for the PCR tests.

15. The outcome of case-control study isn’t the occurrence of diarrhoea. Please revise the measurement of outcomes. Need to mention the measurement for diarrhoea aetiology as well.

Response: 

This has been revised, pages 22 - 23 

The outcome for the case control study would be the occurrence of diarrhoea in a child under five years. This will be used to determine factors associated with the diarrhoea. The outcome for the cross-sectional study will be pathogens causing diarrhoea in the children under five years.

16. There are 2 study population for case-control study (children and mothers/caregivers), you should obtain both assent and consent form.

Response: 

Thank you for your comments. However, in Ghana, children under five years are below the age of assenting, therefore, the caregivers above 18years give consent to be part of the study with their children. The consent given by the caregiver, covers their child under five years being part of the study. 

17. Need to revise the references in line with guidelines of PLOS ONE.

Response 

In-text and end-text references have been reformatted according to the journals’s requirements. 

Thank you

18. English writing still needs to be improved. Thorough proofreading is highly recommended.

Response

Thank you very much for the comment. The entire manuscripts has been taken through copyediting and proofreading now. 

Reviewer #3: Reviewer comments

Abstract

Overall: the Abstract is clear and ok, but some few details are required.

Introduction: You mentioned “decline” – can give us the percentage.

Response 

Page 4, introduction: “From 1990 to 2015, diarrhoea morbidities for have reduced by about 50% (from 1.8 million to 842,000)” 

Methodology: Included the analysis that

---

## [Decision Letter · Decision Letter 2]

9 Apr 2024

A study to evaluate WASH interventions and risk factors of diarrhoea among children under five years, Anloga district, Ghana: a research protocol

PONE-D-23-11119R2

Dear Dr. Bandoh,

We’re pleased to inform you that your manuscript has been judged scientifically suitable for publication and will be formally accepted for publication once it meets all outstanding technical requirements.

Kind regards,

Khin Thet Wai, MBBS, MPH, MA

Academic Editor

PLOS ONE

Additional Editor Comments (optional):

Reviewers' comments:

Reviewer's Responses to Questions

**Comments to the Author**

1. Does the manuscript provide a valid rationale for the proposed study, with clearly identified and justified research questions?

Reviewer #2: Yes

2. Is the protocol technically sound and planned in a manner that will lead to a meaningful outcome and allow testing the stated hypotheses?

Reviewer #2: Yes

3. Is the methodology feasible and described in sufficient detail to allow the work to be replicable?

Reviewer #2: Yes

4. Have the authors described where all data underlying the findings will be made available when the study is complete?

Reviewer #2: Yes

5. Is the manuscript presented in an intelligible fashion and written in standard English?

Reviewer #2: Yes

6. Review Comments to the Author

You may also provide optional suggestions and comments to authors that they might find helpful in planning their study.

Reviewer #2: I do appreciate your effort in revising your manuscript. I've found that you have revised all my comments of previous review.

7. PLOS authors have the option to publish the peer review history of their article (what does this mean?). If published, this will include your full peer review and any attached files.

Reviewer #2: No

---

## [Editor Report · Acceptance letter]

14 May 2024

PONE-D-23-11119R2 

PLOS ONE

Dear Dr. Bandoh, 

I'm pleased to inform you that your manuscript has been deemed suitable for publication in PLOS ONE. Congratulations! Your manuscript is now being handed over to our production team.

Kind regards, 

on behalf of

Dr. Khin Thet Wai 

Academic Editor

PLOS ONE